# Development and initial validation of a simple tool to screen for partner support or opposition to HIV prevention product use

Elizabeth E. Tolley[1]◉*, Seth Zissette[1,2]◉, Andres Martinez[1]◉, Thesla Palanee-Phillips[3]‡, Florence Mathebula[3], Siyanda Tenza[3], Miriam Hartmann[4], Elizabeth T. Montgomery[4]‡

**1** Behavioral, Epidemiological & Clinical Sciences, FHI 360, Durham, North Carolina, United States of America, **2** Notre Dame University, South Bend, Indiana, United States of America, **3** Wits Reproductive Health and HIV Institute, University of the Witwatersrand, Johannesburg, South Africa, **4** Women's Global Health Imperative, RTI International, San Francisco Project Office, California, United States of America

◉ These authors contributed equally to this work.
‡ TPP and ETM also contributed equally to this work.
* btolley@fhi360.org

**Data Availability Statement:** The complete data from the quantitative survey have been uploaded the Harvard Dataverse platform and can be viewed

## Abstract

In HIV prevention trials, male partners have influenced women's ability to adhere to investigational products, including antiretroviral (ARV) containing vaginal rings. Validated scales can be useful tools to systematically measure complex constructs, such as those related to male partner engagement. Although multiple scales exist to assess physical, psychological and sexual violence within intimate relationships, fewer scales focus on supportive behaviors within these relationships. Our intervention involved development of a Healthy Relationship Assessment Tool (HEART) that assessed both positive and negative aspects of male partner involvement in women's HIV prevention. We identified and refined 127 potential items, representing intimate partner violence, agency and social support. A structured survey, including potential items and other sociodemographic and behavioral variables was administered to former microbicide trial and non-trial participants. We conducted an exploratory factor analysis (EFA) to identify a reduced set of constructs and items to screen women who might experience social harms or benefits from vaginal ring use. We examined associations between constructs and with other survey variables to assess content and construct validity. In a subset of 10 women who participated in the survey and qualitative interviews, we used qualitative data to predict survey scores. We retained five constructs with theoretical relevance and good-to-strong reliability for the tool, including: Traditional Values; Partner Support; Partner Abuse & Control; Partner Resistance to HIV Prevention; and HIV Prevention Readiness. Predicted associations between HEART constructs, and correspondence between participants' qualitative data and HEART scores were generally correct, while those between constructs and other sociodemographic variables were more mixed. Initial validation of the HEART tool was promising. The tool will be used during the CHARISMA pilot study at the Johannesburg MTN 025/HOPE site and validated as part of a randomized controlled trial of CHARISMA within a PrEP demonstration project. Beyond clinical trial settings, HEART could assist PrEP or antiretroviral treatment (ART) providers with an

at "https://dataverse.harvard.edu/dataset.xhtml?persistentId=doi:10.7910/DVN/EULT3T." The qualitative data for this paper are fully contained in the paper.

**Funding:** This study was funded by the United States Agency for International Development (USAID) under APS-OAA-14-000076, of which EM was the Principal Investigator. The funders did not play any role in the study design, data collection and analysis, decision to publish or preparation of the manuscript.

**Competing interests:** The authors have declared that no competing interests exist.

easy-to-administer tool to identify risk and tailor risk reduction, empowerment and adherence counseling for microbicides, PrEP or ART related services.

## Introduction

Although the number of people newly infected with HIV has declined over the past two decades [1], young African women continue to be at risk of HIV and are, in fact, one of the few populations that have not benefited from recent declines in HIV incidence [2]. Of the 1.7 million new HIV infections in 2016, almost half (48%) globally were in women, with 59% of infections occurring among young women between 15–24 years of age [3, 4]. In sub-Saharan Africa, 56% of new infections were in women, and 67% of new infections were comprised of young women aged 15–24 [5]. Research suggests that women's continued risk of HIV stems from inequitable gender norms and the threat of intimate partner violence (IPV) that both promote high-risk sexual relationships and limit options for HIV prevention behaviors [6, 7].

Despite the promise of new HIV pre-exposure prophylaxis (PrEP) products, women continue to face challenges related to PrEP adherence. While clinical trials of various regimens of oral PrEP have shown moderate to high levels of protection in HIV-discordant couples and men-who-have-sex-with-men (MSM), several phase 3 trials of oral and vaginal PrEP products among African women failed to show evidence of effectiveness [8]. Across these trials, product adherence has been reportedly low [9–11]. A number of factors have been associated with low or intermittent adherence including low perceived risk perception [12], ambivalence towards research and/or use of investigational products [13], and challenges related to managing social relationships [14–16]. Women's approaches to engaging their male partners in their HIV prevention product use and trial participation have ranged from open communication to gradual disclosure to entirely covert use [17]. Indeed, in some cases, participating in an HIV prevention trial has even enhanced couple communication [18]. Non-disclosure has been attributed either to fearing a partner's negative and potentially violent reaction or to a belief that the decision for PrEP use and/or trial participation is the woman's alone to make [17]. Evidence from HIV prevention and treatment trials have reported improvements in adherence among those who have disclosed [19, 20], but challenges with disclosure to partners may impede adherence [20, 21]. And, while reports of social harms, including intimate partner violence (IPV) have been relatively infrequent in PrEP trials, women who reported recent IPV had a higher risk of low adherence than those who reported no IPV or IPV episodes that occurred in the more distant past [22, 23].

Given male partners' variable role in either enhancing or inhibiting women's use of HIV prevention products, there is a need for simple screening tools that could help trial implementers or indeed PrEP providers to better tailor PrEP counseling and adherence support to women's sexual partner contexts. Nevertheless, to our knowledge, no tools exist to date that aim to assess the full spectrum of partner engagement, from supportive to abusive. Our study set out to develop such a tool.

## Materials and methods

We developed a Social Benefits-Harms Tool, later referred to as the Healthy Relationship Assessment Tool or HEART, as a preliminary step in the development of a couples'-based HIV prevention intervention called the Community Health Clinic Model for Agency in Relationships and Safer Microbicide Adherence Pilot study, or CHARISMA. Validated scales can

be useful tools to systematically measure complex constructs [24, 25], such as male partner engagement. Therefore, the HEART was conceptualized as a set of brief scales to measure aspects of a woman's own agency, her attitudes towards more traditional gender norms that might dampen this agency, and the degree to which her primary sexual partner was perceived as supportive, controlling or violent. Within the CHARISMA intervention, the HEART was intended to assist lay counselors in determining which of several counseling modules and/or referral mechanisms should be offered based on the potential benefits or harms a participant might encounter from using the vaginal ring. (This tool, as a component of the CHARISMA behavioral counseling intervention, would be piloted within the MTN-025/ HOPE trial of the dapivirine vaginal ring.) In this paper, we describe the development, including finalization of construct and item content, and initial validation of the tool.

## Project design and objectives

The goal of this study was to develop and provide initial validation for a set of scales that could be used within a clinical context to identify aspects of a woman's primary relationship that might facilitate or inhibit her consistent use of an HIV prevention product. We followed a standard scale development process [25]. This entailed identifying a pool of potential items and underlying constructs that would be theoretically associated with our intended measure; conducting rounds of cognitive interviews to ensure that potential items were salient, easy to understand and to respond to; administering a survey with the refined items and other potentially associated variables to enable exploratory factor analysis and psychometric evaluation of the resulting constructs; and using qualitative data to predict survey scores in a subset of women to further assess construct validity.

## Theoretical framework for HEART

As described elsewhere [26], we reviewed the scientific and gray literature to identify validated scales or other tools that measured various aspects of intimate relationships [27], including agency [28], social support, as well as partner abuse and violence [29]. We gave precedence to scales or tools used in African contexts. IPV measures generally took one of two approaches: observable or reportable physical behaviors (e.g., hitting, pushing, or kicking), or perceptions of psychological abuse (e.g., embarrassing, withholding financial support, or blaming). Many scales contained a battery of 15–40 items assessing perpetration of violent acts (ever or at stated intervals) by men against women; some were bi-directional (e.g., also women against men). A few tools, such as the four-item "HITS" questionnaire (Hurt, Insulted, Threatened with harm, and Screamed), were much shorter and meant to be administered in clinic settings [30]. Fewer social support measures were identified, and those we found usually measured situation-specific types of social support (e.g., partner supportive behaviors during the prenatal period) [31]. Our review confirmed our initial hypothesis that a new tool would be needed to represent both the supportive as well as abusive or harmful dimensions of an intimate relationship without being overly time consuming to administer.

Based on the literature review, we identified 135 potential items for the HEART representing agency, social support or IPV. Items were translated into isiZulu. We conducted three rounds of cognitive interviews with a total of 25 women with and without prior experience participating in clinical trial research and/or past experiences of violence to assess ease, comprehensibility, and relevance of items. (Women's eligibility was the same as described below for the HEART survey.) Interviews were conducted in either English or isiZulu, or a mix of both languages. By the end of the third round of cognitive interviews, eight items that were

reported to be embarrassing or duplicative had been removed, and some items that had unclear wording or difficult answer choices were edited.

## HEART survey procedures

Between April and September 2016, we administered a cross-sectional survey to 309 women in Johannesburg, South Africa. Eligible participants were aged 18–40, residing in communities where previous HIV prevention trials (including the MTN-020/ ASPIRE vaginal ring trial [11], the MTN-003/VOICE vaginal gel and oral PrEP trial [10] and the FACTS vaginal gel trial) [32] had been conducted, not currently participating in other related research studies, and willing and able to provide written informed consent. We recruited both former HIV prevention trial participants (FTPs) who had given permission to be contacted at study exit for future research and women recruited from adjacent non-research public clinics (trial-naïve participants, TNP) through staff led recruitment efforts. The survey included 127 theoretically-relevant items, refined through the cognitive interview process. Responses for most potential HEART items were based on a 6-point scale ranging from "1 = Disagree A Lot" to "6 = Agree A Lot". In addition, we included sociodemographic and self-reported risk behavior variables. The survey was administered on tablets by trained social scientists in a private location in the Hillbrow Wits RHI clinic. Participants had the option to take the survey in English, isiZulu, or to switch between languages for different items.

Prior to implementation, the study was reviewed and approved by the FHI 360 Protection of Human Subjects Committee and the Wits Human Research Ethics Committee in Johannesburg. Written consent was obtained from each participant in their preferred language of English or Zulu.

## Survey analyses

Our initial goal was to recruit a minimum of 400 survey participants with a minimum of 250 former trial participants and 150 similar community members. However, due to a shorted recruitment time and lower than expected access to former—but not current trial participants, we terminated survey recruitment after enrolling 309 participants.

**Item and dimension analysis.** Following survey completion, we examined the response distribution, means, standard deviations, skewness and kurtosis of all items. We first eliminated four highly skewed items having a mean score below 1.4 or above 5.8, as the floor and ceiling effects demonstrated by these items produced virtually no variability between participants. We then conducted an exploratory factor analysis (EFA) including all non-excluded scale items to determine the number and content of underlying domains for the HEART and to reduce the overall number of items that represented each factor. We followed standard EFA procedures, including use of Promax rotation (allowing for cross-loading of items on multiple factors) and retention of items with a loading of 0.4 or higher on a resulting factor. We examined the eigenvalues and Cattell's scree plot to determine the number of factors to extract, and the item-to-total and inter-item correlations to determine which items to retain. We assessed the internal consistency of our resulting factors, deleting any items that did not maintain or increase a factor's Cronbach's alpha. (Note: Our final solution contained only two items that had similar and relatively low cross-loadings. We retained the two items on the factor based on our analysis of internal consistency.)

**Initial construct validation.** We assessed construct validity in several ways. First, based on the resulting factors, four geographically distinct groups within our project team developed separate *a priori* predictions about the direction and strength of correlations (i.e., significant positive association, significant negative association or non-significant) between the potential

scales and 10 relevant sociodemographic and behavioral variables, as well as among the resulting scales themselves. Our four groups included two US-based FHI 360 team members, two US-based RTI members, one SA-based RTI member and three SA-based Wits RHI members. Each team submitted their predictions which were tabulated prior to analysis. We then examined the associations between the factors and the theoretically relevant variables included in our survey, using Pearson product-moment correlation statistics. We considered evidence of "as-hypothesized" relationships to provide initial validation that our tool is performing as needed.

### Parallel qualitative interviews

Qualitative, in-depth interviews (IDIs) were conducted with 10 survey participants to obtain input on acceptability, feasibility and perceived effectiveness of intervention components proposed for CHARISMA. As many of the topics discussed in interviews thematically related to domains (and often, specific items) on the HEART, we used participants' IDI data to make predictions about their scale scores on HEART and to determine whether any salient constructs pertaining to the influence of partner dynamics on PrEP use were missing from the tool.

**IDI analysis.** Transcripts were thematically coded in NVivo by three independent coders, following a basic codebook that included the five HEART constructs and, within each construct, "high", "medium" and "low" sub-codes to indicate whether the text segment would predict a high, medium or low score on the construct. The coders double-coded one of the ten interviews and met to determine whether they had applied the same text segments to the five broad HEART codes, as well as specific sub-codes. Differences were reconciled through discussion between all coders to standardize the use of codes during the coding process. For each of the 10 IDI participants, coders then examined the frequency and content of text segments to make predictions about whether they would score "high" ($\geq 1$ standard deviation (sd) above the mean), "medium" (within 1 sd from the mean), or "low" ($\leq 1$ sd below the mean) on each scale. For each of the five constructs, coders assessed the number of correct predictions out of a possible 10 participant scores.

As a second level of analysis, a codebook containing codes for each item in the HEART was developed and applied to specific items within the tool. This step was used to identify any noteworthy concepts that arose consistently across participants but were not captured by an existing item. We analyzed the rates at which existing items in the HEART were reflected by natural discussion in the interview transcripts, identified items that were not reflected by any discussion and any potential new items that were discussed but were not currently a part of the HEART.

## Results

### Sociodemographic information

Most participants in our survey (79%) had not previously participated in HIV prevention clinical research (Table 1). Those with prior trial experience were most likely to have taken part in the MTN-020/ ASPIRE vaginal ring trial (88%), although some participated in trials for vaginal gels or other products. FTPs were, on average, several years older than TNP, more likely to be living with a partner and more likely to earn an income.

As presented in Table 2, less than a third of participants (28%) were sure that their partners did not have sexual partners and most participants (72%) reported having discussed condom use with their partner at last sex. Very few participants (8%) reported feeling persuaded or coerced into having sex during their most recent sexual encounter.

**Table 1. Socio-demographic information of surveyed participants, overall and by trial experience subgroup.**

| | Total(n = 309) | TNP(n = 245) | FTP(n = 64) |
|---|---|---|---|
| Years of age (mean) | 27 | 26* | 29* |
| | % | % | % |
| Ever participated in an HIV prevention clinical trial | 21 | n/a | 100 |
| Type of trial: | | | |
| Vaginal ring | | | 88 |
| Vaginal gel | | n/a | 5 |
| Other (not oral PrEP) | | | 8 |
| Type of residence: | | | |
| Free-standing or town house | 33 | 33 | 30 |
| Flat | 44 | 45 | 42 |
| Single room | 15 | 14 | 19 |
| Informal or other housing | 9 | 9 | 9 |
| Currently living with: | | | |
| Parent(s) (mother and/or father) | 25 | 26 | 22 |
| Other family (non-parent) | 29 | 29 | 28 |
| Primary sex partner | 21 | 18* | 30* |
| Own children | 26 | 24 | 30 |
| Other | 28 | 31* | 17* |
| Have children | 68 | 66 | 77 |
| Number of people living in household (mean) | 4 | 4 | 5 |
| Highest level of education: | | | |
| Primary or less | 2 | 1 | 2 |
| Secondary, not complete | 26 | 27 | 22 |
| Secondary, complete | 37 | 34 | 48 |
| Some college or university | 24 | 27 | 14 |
| College or university, complete | 12 | 12 | 14 |
| Earns an income | 36 | 32* | 52* |

*p < .05.

## Analysis of items and conceptual dimensions

We began with 127 potential items and ultimately retained 42 loading onto five factors (Table 3). The final five-factor solution was originally derived from a four-factor solution in which one factor contained a large number of items (28), resulting in an artificially inflated measure of internal reliability. We then conducted a subsequent EFA with this factor, resulting in three sub-factors. We retained these three factors along with two of the remaining factors from the original four-factor solution. The final remaining factor from the original four-factor solution was dropped due to lack of face validity. Overall, the number of missing responses across all items was small. In fact, all but one of the 42 items had at least 305 valid responses, out of a potential for 309. The only item with a substantial amount of missingness was item 5 in the Partner Abuse & Control scale, with only 277 valid responses. While missingness was uncommon, ceiling and flooring effects—demonstrated by limited variation in responses and/ or high levels of skewness in the item responses distributions—were common and in some cases substantial. For example, more than half the responses to all the items in the Partner Resistance to HIV Prevention were in the "Disagree a Lot" response option (lowest). Similarly, more than half the responses to all the items in the HIV Prevention Readiness scale were in the

**Table 2. Sexual behavior characteristics of surveyed participants, overall and by trial-experience subgroup.**

|  | Total(n = 309) | TNP(n = 245) | FTP(n = 64) |
|---|---|---|---|
|  | % | % | % |
| Current relationship status: |  |  |  |
| Married or living with regular partner | 28 | 24* | 42* |
| Regular partner(s), not living together | 67 | 71* | 52* |
| Sexually active, no regular partner(s) | 2 | 2 | 3 |
| Not sexually active currently | 4 | 4 | 3 |
| *Has regular sex partner (married, living with or not)* | *(n = 291)* | *(n = 231)* | *(n = 60)* |
| Years with current/regular partner (mean) | 5 | 4 | 5 |
| With current/regular partner in last clinical trial |  | n/a | 78 |
| Regular partner provides financial or material support | 85 | 84 | 90 |
|  | *(n = 309)* | *(n = 245)* | *(n = 64)* |
| Age of main sex partner (mean) | 31 | 30* | 34* |
| Partner(s) have other sex partners: |  |  |  |
| No | 29 | 30 | 25 |
| Don't know | 51 | 48 | 63 |
| Yes | 20 | 22 | 13 |
| Discussed condom use at most recent sex act | 72 | 71 | 75 |
| Used a condom at most recent sex act | 56 | 58 | 50 |
| Willingness to have sex at most recent sex act: |  |  |  |
| Willing or wanted to have sex | 93 | 91 | 97 |
| Persuaded or talked into sex | 7 | 8 | 3 |
| Coerced or threatened | 1 | 1 | 0 |
| Physically held down or raped | 0 | 0 | 0 |

*p < .05.

"Agree a Lot" response option (highest). Some of the items in the other scales also exhibited limited variability and high levels of skewness (see S1 Table). The flooring and ceiling effects limit the ability of the scales and of the tool more generally to distinguish 'average' respondents from one another.

From a conceptual perspective, the 5-factor solution met the aims of our tool development. It included a 13-item "Traditional Values" construct that elicited attitudes about gender roles and women's agency to make decisions; a 10-item "Partner Support" construct that included a mix of statements about a partner's support or lack of support for the relationship; a 9-item "Partner Abuse & Control" construct that included statements about a partner's physical and emotionally abusive behaviors and two items indicating a woman's lack of control over life decision; and two 5-item factors, the first indicating "Partner Resistance to HIV Prevention" and the second indicating "HIV Prevention Readiness". Internal reliability of the items was good (Cronbach's Alpha 0.80 or higher) for four of the five factors and borderline acceptable for the HIV Prevention Readiness factor (0.68).

## Initial construct validation

We provide an example of our independent predictions between the Partner Abuse & Control scale and the other HEART scales in Table 4. Each group's predictions are stated and indicated by a color (gray for neutral or non-significant, blue for positive and red for negative

**Table 3. Exploratory factor analysis 5-factor solution for development of HEART.**

| Factor and items | Item loading |
|---|---|
| **Factor 1: Traditional Values** *(a = .84, Scale range[+] 13–78)* | |
| 1. Changing diapers, giving the kids a bath, and feeding the kids is a mother's responsibility. | 0.65 |
| 2. I think that a woman cannot refuse to have sex with her husband. | 0.62 |
| 3. I think that if a man has paid lobola for his wife, he owns her. | 0.61 |
| 4. A woman should always listen and abide by the word of her husband without questions. | 0.60 |
| 5. A man should have the final word about decisions in his home. | 0.59 |
| 6. A real man produces a male child. | 0.58 |
| 7. I think that if a man has paid lobola for his wife, she must have sex when he wants it. | 0.57 |
| 8. I think that a man should have the final say in all family matters. | 0.46 |
| 9. A woman should accept her partner's wishes—even when she disagrees—to keep the family together. | 0.46 |
| 10. I only think I am attractive if other people think I am. | 0.45 |
| 11. I think that there is nothing a woman can do if her husband wants to have girlfriends. | 0.42 |
| 12. If someone insults a man, he should defend his status with force if he has to. | 0.42 |
| 13. It's only rape if a woman fights back. | 0.41 |
| **Factor 2: Partner Support** *(a = .81, Scale range[+] 10–60)* | |
| 1. My partner is as committed as I am to our relationship. | 0.65 |
| 2. I feel comfortable telling my partner that I see things differently. | 0.48 |
| 3. In general, my relationship has a lot of tension. (*reverse scored*) | 0.59 |
| 4. It takes a long time to work out arguments with my partner. (*reverse scored*) | 0.59 |
| 5. I feel trapped or stuck in our relationship. (reverse scored) | 0.58 |
| 6. Arguments with my partner result in me feeling down or bad about myself. (*reverse scored*) | 0.47 |
| 7. My partner does what he wants, even if I do not want him to. (*reverse scored*) | 0.42 |
| 8. I feel safe in my current relationship. | 0.59 |
| 9. My partner takes my earning or refuses to give me money when he has money for other things. (*reverse scored*) | 0.48 |
| 10. My partner is/will be very supportive of my use of an HIV prevention product. | 0.42 |
| **Factor 3: Partner Abuse & Control** *(a = .81, Scale range[+] 9–54)* | |
| 1. My partner slaps, hits, kicks, or pushes me. | 0.79 |
| 2. My partner chokes, pulls hair, or burns me. | 0.77 |
| 3. My partner does things to scare or intimidate me on purpose. | 0.69 |
| 4. My partner makes fun of me or humiliates me. | 0.62 |
| 5. My partner makes most of the decisions about how the household finances are used. | 0.54 |
| 6. I feel frightened by what my partner says or does. | 0.48 |
| 7. My partner won't allow me to wear certain things. | 0.43 |
| 8. I can't seem to make good decisions about my life. | 0.47 |
| 9. I do not trust myself to make good decisions about my life. | 0.41 |
| **Factor 4: Partner Resistance to HIV Prevention** *(a = .80, Scale range[+] 5–30)* | |
| 1. If I asked my partner to use a condom, he would get angry. | 0.76 |
| 2. If I asked my partner to use a condom, he would think I'm having sex with other people. | 0.70 |
| 3. If I asked my partner to use a condom, he would get violent. | 0.69 |
| 4. I cannot tell my partner about HIV prevention product use because he will become angry. | 0.61 |
| 5. If I asked my partner to use an HIV prevention product, he would get violent. | 0.61 |
| **Factor 5: HIV Prevention Readiness** *(a = .68, Scale range[+] 5–30)* | |
| 1. I understand the risks and benefits of HIV prevention product use and have chosen to use them. | 0.52 |
| 2. Using an HIV prevention product is the right thing to do. | 0.58 |
| 3. Using an HIV prevention product shows that my partner and I care about each other. | 0.63 |

(*Continued*)

**Table 3.** (Continued)

| Factor and items | Item loading |
|---|---|
| 4. Using an HIV prevention product with my partner will help us communicate better. | 0.54 |
| 5 HIV prevention products would help me protect myself. | 0.41 |

[+] Factor scores were calculated as the sum of the item responses; these ranges reflect that sum. The average available item response was imputed for any missing values.

correlation). Correlations are bolded if they were predicted in both direction and magnitude by at least three of the four project team groups.

As illustrated, our predictions were generally correct for correlations between the Partner Abuse & Control scale and three of the other scales. However, we were less successful at predicting the magnitude of associations with the sociodemographic variables. For example, fewer than three of the four groups predicted that Partner Abuse & Control was both significantly and negatively associated with education level, past trial participation or being with the same partner who was with them in a past trial. In a similar way, groups more often predicted strong, negative correlations between Partner Abuse & Control and behaviors such as discussing and/or using a condom at last sex, when the correlation was insignificant.

Overall, at least one group correctly predicted the direction and magnitude of the correlations between the scales and with other variables 83% of the time. The ability to correctly predict correlations differed by factor, from at least one group making correct predictions 93% of the time for Partner Abuse & Control to just 71% of the time for HIV Prevention Readiness.

Fig 1 shows the distribution of individual scale scores (down the main diagonal) and the correlation and significance between each pair of scales (top-right). The bottom-left cells

**Table 4. Predicted correlations between Partner Abuse & Control and other scales and socio-demographic variables.**

| Variables | Correlation | Predicted Direction | | | |
|---|---|---|---|---|---|
| | | Group A | Group B | Group C | Group D |
| Other HEART scales | | | | | |
| **Traditional Values** | **0.41**[*] | Neutral | **Positive** | **Positive** | **Positive** |
| **Partner Support** | **-0.64**[*] | **Negative** | **Negative** | **Negative** | **Negative** |
| **Partner Resistance to HIV Prevention** | **0.53**[*] | **Positive** | **Positive** | **Positive** | **Positive** |
| HIV Prevention Readiness | -0.09 | **Neutral** | Negative | Negative | Negative |
| Socio-demographic variables | | | | | |
| Age | **0.13** | **Neutral** | **Neutral** | Negative | **Neutral** |
| Education | -0.39[*] | Neutral | Neutral | **Negative** | Neutral |
| Past clinical trial participation | -0.19[*] | Neutral | Positive | **Negative** | **Negative** |
| Time with partner | 0.16[*] | Neutral | Neutral | Neutral | Neutral |
| Current partner, last clinical trial | -0.48[*] | Neutral | Neutral | **Negative** | **Negative** |
| **Partner provides material support** | -0.12 | Positive | **Neutral** | **Neutral** | **Neutral** |
| Discussed condom use at last sex | -0.01 | **Neutral** | Negative | Negative | Negative |
| **Coerced at last sex** | **0.24**[*] | **Positive** | **Positive** | **Positive** | **Positive** |
| Used a condom at last sex | -0.15 | **Neutral** | Negative | Negative | Negative |
| Earns own income | -0.09 | **Neutral** | Negative | Negative | **Neutral** |

[*]Statistically different from 0 at a 0.05 Type I error level.

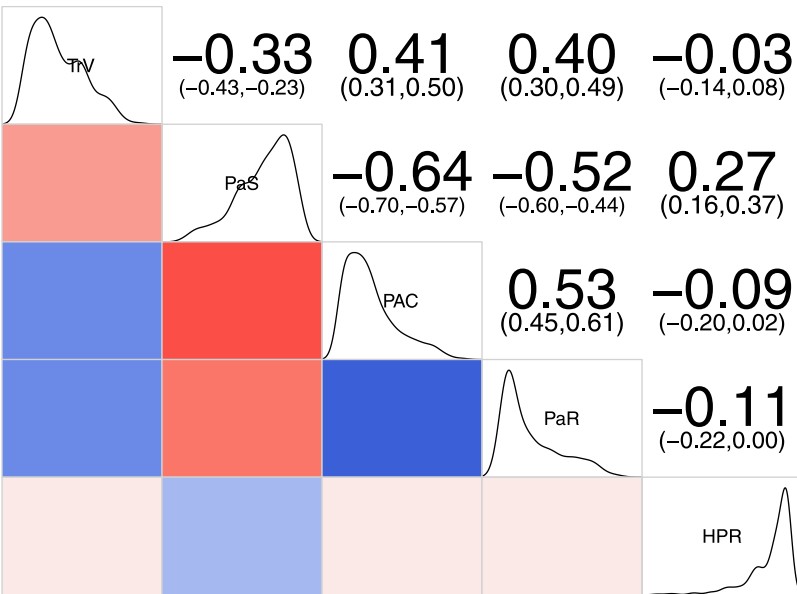

**Fig 1. Distributions of and correlations among HEART scales.** TrV: Traditional Values; PaS: Partner Support; PAC: Partner Abuse & Control; PaR: Partner Resistance; HPR: HIV Prevention Readiness. Upper-right panels contain the Pearson correlation coefficient with the 95% confidence interval in parenthesis. Lower-left panels contain a visual cue for the direction and magnitude of the correlation coefficient.

visually indicate the direction (positive = blue; negative = red) of the correlation, with the magnitude (low, medium and high) depicted through the shading of each color. We note that Partner Resistance to HIV Prevention was strongly positively skewed and HIV Prevention Readiness strongly negatively skewed. As predicted, Partner Support was negatively associated with Traditional Values (-0.33), Partner Abuse & Control (-0.64), and Partner Resistance to HIV Prevention (-0.52). Also, HIV Prevention Readiness exhibited a positive association with Partner Support (0.27), but a weak association with the other scales (Fig 1).

## Comparisons between qualitative text and HEART scores

In Table 5, we provide illustrative quotes, the resulting prediction made by coders and the rationale for making the prediction related to two of the HEART scales (Partner Abuse & Control and HIV Prevention Readiness). During qualitative interviews, women could describe their partner relationships in detail, providing multiple and sometimes contradictory examples of support, control, abuse and how such interactions affected HIV prevention decision-making. Accurate predictions of HEART scales scores based on these more nuanced descriptions would provide additional evidence for each scale's construct validity.

Table 6 presents the number of text segments for each participant that were coded as low, medium or high for two HEART domains, as well as our final score range prediction, and the actual score range for that participant. (Illustrative quotes and predicted versus actual scores for all scales are available in the supplemental materials.) Predictions were highly accurate for most of the scales, at 80% for the Partner Abuse & Control, Traditional Values, and Partner Support scales and 70% for the Partner Resistance to HIV Prevention Scale. The final scale, HIV Prevention Readiness, experienced greater discrepancy, with only 30% of predictions based on thematic analysis matching actual scores. Our ability to predict the strength and

**Table 5. Illustrative quotes from IDIs used to predict HEART score ranges for Partner Abuse & Control and HIV Prevention Readiness scales.**

| ID | Score Prediction | Partner Abuse & Control Example Quotes | Rationale |
|---|---|---|---|
| 1007 | Low | INTERVIEWER: Can you describe a time when your partner made you feel scared or humiliated?<br>PARTICIPANT: Nothing. I didn't feel humiliated or scared. Unless, [when] it was April Fool's day. He lied that he got into an accident, stuff like that and I was so shocked. And then after fifteen minutes, no, don't you know that it's April Fool's day. | Participant experiences support and trust in her relationship and denies that partner has ever abused her physically or emotionally. |
| 1026 | Medium | INTERVIEWER: When he teased you about your weight loss, how did you feel?<br>PARTICIPANT: Irritated, I hated every minute of it. I hated it.<br>INTERVIEWER: Did you feel humiliated?<br>PARTICIPANT: Ja in a way, I mean why would you? it's not funny and he knew I was complaining that I lost a lot of weight. I don't even want to bring up the conversation about my weight, because I hate it.<br>INTERVIEWER: Can you describe a time when you were worried your partner might be or was ever violent?<br>PARTICIPANT: No, never. I'm almost sure he can't raise a hand. | Participant has experienced some humiliation from partner when teasing her about her weight but has never experienced physical abuse. |
| 1012 | High | PARTICIPANT: He did it once. Checked my WhatsApp. I felt nervous, because I was chatting with my ex [boyfriend]. Then he asked me, 'Why are you still WhatsApping with your ex?'<br>INTERVIEWER: Hmm.<br>PARTICIPANT: And so, he took my phone and showed me the messages. So, I said sorry. Ja. Like I'm making him feel, he beat me up. So, I said sorry, but I can't stop chatting with him, because it's Whatsapp. It's not like maybe if you broke up with somebody you don't chat with them. No, you are wrong on that.<br>NTERVIEWER: Okay. And what did he do?<br>PARTICIPANT: He beat me with a belt buckle. | Participant experiences abuse due to her partner's jealousy. |
| ID | Score Prediction | HIV Prevention Readiness Example Quotes | Rationale |
| 1009 | Low | PARTICIPANT: Ja, I was a bit worried continuing using the ring. As I've said, like I had a problem with infections, thinking maybe it was the ring. Ja, it was a problem until I stopped using it. | Participant was worried about ring use and the risks associated with it. |
| 1007 | Medium | PARTICIPANT: No. I didn't feel that I was at risk of getting HIV. But the other thing that made me nervous because of we, in the family we have people who died with AIDS, but we're having others who have lived a long time with AIDS. So that's the other thing that made me to become nervous. | Participant did not feel at risk herself but decided to participate after reflecting on family members who have died of AIDS. |
| 1026 | High | INTERVIEWER: How could use of an HIV prevention product make you feel closer to your partner?<br>PARTICIPANT: It can definitely improve on being closer to their partner because there's no fear.<br>INTERVIEWER: Fear of what?<br>PARTICIPANT: Even if you know that, chances are, he's cheating, he's doing stuff, but you are not unprotected, hence there's no fear. You just let yourself be. | Participant believes that use of an HIV prevention product can improve relationship with partner. |

direction of scores on the Traditional Values, Partner Support, Partner Abuse & Control, and Partner Resistance to HIV Prevention scales supports the construct validity of these four.

Finally, for most items in the five HEART scales, we identified multiple matching examples of text segments in our qualitative data. All items in the Partner Support scale were represented by quotes from at least one and as many as all 10 IDI participants, while only three of the five items in the Partner Resistance to HIV Prevention scale were represented by similar qualitative text segments. The item-level qualitative analysis also identified other potential items that fit within the five HEART constructs but didn't clearly match existing items. For example, some participants identified concerns about trial procedures (e.g., HIV tests and other clinic procedures), or product-specific concerns (e.g., worry that the vaginal ring will affect sex life) as

**Table 6. Predictive text frequencies and comparison of predictions to survey score range, by IDI.**

| ID | Low Score References (%) | Medium Score References (%) | High Score References (%) | Overall Score—Prediction | Overall Score—Actual | Correct? |
|---|---|---|---|---|---|---|
| PARTNER ABUSE & CONTROL PREDICTIONS | | | | | | |
| 1006 | 8 | 59 | 33 | Medium | Medium | Yes |
| 1007 | 65 | 35 | 0 | Low | Low | Yes |
| 1009 | 33 | 67 | 0 | Medium | Low | No |
| 1011 | 100 | 0 | 0 | Low | Low | Yes |
| 1012 | 8 | 14 | 78 | High | High | Yes |
| 1018 | 0 | 0 | 100 | High | Medium | No |
| 1019 | 24 | 76 | 0 | Medium | Medium | Yes |
| 1023 | 22 | 78 | 0 | Medium | Medium | Yes |
| 1026 | 7 | 85 | 9 | Medium | Medium | Yes |
| 1032 | 6 | 94 | 0 | Medium | Medium | Yes |
| HIV PREVENTION READINESS PREDICTIONS | | | | | | |
| 1006 | 38 | 41 | 21 | Medium | High | No |
| 1007 | 32 | 54 | 14 | Medium | High | No |
| 1009 | 13 | 27 | 60 | High | Low | No |
| 1011 | 0 | 28 | 72 | High | Medium | No |
| 1012 | 0 | 20 | 80 | High | Medium | No |
| 1018 | 41 | 59 | 0 | Medium | Medium | Yes |
| 1019 | 21 | 0 | 79 | High | High | Yes |
| 1023 | 0 | 49 | 51 | High | High | Yes |
| 1026 | 0 | 0 | 100 | High | Medium | No |
| 1032 | 0 | 27 | 73 | High | Medium | No |

discouraging their readiness to use HIV prevention products. Table 7 provides examples of illustrative quotes for seven of the nine items retained on the Partner Abuse & Control scale. (Examples for existing and proposed items for other scales can be found in the supplemental materials.)

## Discussion

These analyses, based on data from the scale survey used to develop the HEART tool, support overall validity of four of the five HEART scales. Retained factors and composition of items on each scale made theoretical sense and were confirmed by the moderate to strong internal reliability of all but the HIV Prevention Readiness scale.

Additionally, evidence for construct validity of four of the five scales was shown by our quantitative correlation analysis. All but the HIV Prevention Readiness scale correlated in predictable ways with other scales. Our project groups had less agreement on the magnitude of correlations between scales and other sociodemographic and psychosocial variables. It is possible that responses to some of these variables (e.g., discussed or used a condom at last sex) were themselves influenced by social desirability bias. Nevertheless, our predictions were more often than not in the correct direction. In addition, our ability to make accurate predictions about how women scored (low, medium or high) on four of the five HEART scales based on their qualitative interviews, as well as the identification of text segments in multiple IDIs that match scale items provide further support for HEART scales and items.

However, our analysis also indicates some ways that the tool could be strengthened. Overall, women tended to favor extreme values (either 1 or 6) on most items. This may be, in part,

**Table 7. Examples of specific Partner Abuse & Control scale items reflected in the qualitative IDIs.**

| Item | # of Transcripts | Example Quote |
|---|---|---|
| 1. My partner slaps, hits, kicks, or pushes me. | 6 | INTERVIEWER: How often did it happened that he slaps up? PARTICIPANT: It happened once because I drank alcohol without his permission. He slapped me. |
| 2. My partner chokes, pulls hair, or burns me. | 1 | INTERVIEWEE: Yah, he pulled me with the hair, and the twist fell down, like it was miserable, my uncles were not there and my brothers were not there. . . ..no one was at home, it was just my little sisters. . ..they came out. . .I started shouting and my mother called him asked him "what is wrong with you and why are you abusing my child so much |
| 3. My partner does things to scare or intimidate me on purpose. | 3 | INTERVIEWER: Okay. How did you respond when your partner said if you cheat on him, he will kill you? PARTICIPANT: I said, "Kill me", I told him that, "You can kill me. If you want to kill me kill me, I'm not scared of you". |
| 4. My partner makes fun of me or humiliates me. | 3 | INTERVIEWER: Another difficult thing is when men are frightening or saying humiliating things to women, can you describe a time when your partner made you feel scared or humiliated? PARTICIPANT: I know he used to just tease me about my weight, because I lost a lot of weight as compared to when we met, like it was irritating, it was annoying me, just that. |
| 5. My partner makes most of the decisions about how the household finances are used. | 8 | INTERVIEWER: Financial decisions and control can be another difficult thing to negotiate with a partner. How do you and your partner deal with finances? PARTICIPANT: When it comes to the money, I don't have much to say because I never ask him 'How much do you get paid?' because he usually gets paid every week. . . But I do get money to eat, money to buy clothes. Actually, I do not normally buy clothes. The only thing that I do, I use an account. So, I don't have much to say when it comes to money. |
| 6. I feel frightened by what my partner says or does. | 2 | PARTICIPANT: My partner made me feel scared one day because he told me that if I could cheat on him, he would kill. So, I told my mom "Mom, he said if I could cheat on him he would kill me." I was nervous asking myself "What would he kill me with?" |
| 7. My partner won't allow me to wear certain things. | 2 | PARTICIPANT: When I visited my family or relatives. When I'm wearing leggings or something short, he would say "Can't you see that you are have undressed?" So, I feel he is controlling on such things. |
| 8. I can't seem to make good decisions about my life. | 0 | N/A |
| 9. I do not trust myself to make good decisions about my life. | 0 | N/A |

culturally reinforced [33, 34]. For example, women may find it difficult to keep in mind gradations of agreement and use the full range of response options. If so, it is not clear whether extending the response scale—for example, using a 10-point scale rather than a six-point, would alleviate these floor and ceiling effects. On the other hand, the extreme values of some of our items could have resulted from tapping into socially desirable norms. This would suggest rewording items in ways that make them more difficult to fully (dis)agree with.

Finally, the HIV Prevention Readiness scale did not perform well, compared to the other four scales. The internal reliability of the scale is relatively weak, falling short of the .7 cut-off. In addition, items on the scale show the highest skew and kurtosis of all the constructs. Finally,

it is possible that the generic use of "HIV prevention product" rather than a specific product (e.g., vaginal ring) and the absence of items measuring readiness to undergo clinic procedures or navigate product-specific issues explain the scale's lack of correlations with other predicted measures. The scale could potentially be improved by adding new, cognitively tested items, perhaps identifying them from the qualitative interviews.

The strongest evidence for HEART validity would be its ability to discriminate between women with different levels of relationship harmony, stress or violence and to reliably track changes in relationship context over the duration of their participation in a behavioral intervention. To this end, the HEART was included as an intervention component, in our CHARISMA pilot intervention study within the Hillbrow, Johannesburg site of the MTN-025 HOPE vaginal ring trial [26]. The HEART guided recommendations for provision of counseling content at baseline, and was administered at baseline, Month 3 and Month 6 to measure change in relationships over time. The performance of HEART in the CHARISMA pilot, including how changes in HEART scores are associated with receipt of specific counseling modules by pilot study participants, will be described separately. Further, the HEART is currently incorporated within the CHARISMA intervention that is being tested at the same research site using a randomized, controlled design to assess its effect on improving PrEP adherence and partner support and communication; and reduction of IPV.

## Conclusion

We set out to develop and validate a simple tool to assess women's relationships and risk for harmful gender norms and IPV to best address challenges to HIV prevention use that stem from partner relationships. Initial validation of the HEART is promising. Further evidence on the tool's ability to characterize women's relationships, recommend beneficial counseling content and monitor relationship changes over time within several product use contexts is forthcoming. The availability of such a tool to screen for and monitor the impact of women's sexual relationships on their use of HIV prevention products could support providers and programs to better address partner-related challenges that undermine PrEP uptake and adherence.

## Supporting information

**S1 Table. HEART Item descriptive statistics: Number of observations (n), mean, standard deviation (SD), median, minimum (min), maximum (max), skewness, kurtosis, and percentage of responses in response categories 1 through 6.**
(DOCX)

**S2 Table. Illustrative quotes from IDIs used to predict HEART score ranges for Traditional Values, Partner Support, and Partner Resistance to HIV Prevention scales.**
(DOCX)

**S3 Table. Predictive text frequencies and comparison of predictions to survey score range, by IDI.**
(DOCX)

**S4 Table. Examples of specific scale items reflected in qualitative interviews for Traditional Values, Partner Support and Partner Resistance to HIV Prevention.**
(DOCX)

## Acknowledgments

We would like to acknowledge the willing contributions of the women who participated in this study without whom this study would not have been possible. The contributions of the CHA-RISMA study team and community stakeholders are acknowledged as critical in the implementation of this study. This program is made possible by the generous assistance from the American people through the U.S. President's Emergency Plan for AIDS Relief (PEPFAR) and the U.S. Agency for International Development (USAID). The contents are the responsibility of the authors and do not necessarily reflect the views of PEPFAR, USAID or the United States Government.

## Author Contributions

**Conceptualization:** Elizabeth E. Tolley, Elizabeth T. Montgomery.

**Data curation:** Elizabeth E. Tolley, Seth Zissette, Florence Mathebula.

**Formal analysis:** Elizabeth E. Tolley, Seth Zissette, Andres Martinez.

**Funding acquisition:** Elizabeth E. Tolley, Thesla Palanee-Phillips, Elizabeth T. Montgomery.

**Investigation:** Thesla Palanee-Phillips, Florence Mathebula, Siyanda Tenza, Miriam Hartmann.

**Methodology:** Elizabeth E. Tolley, Elizabeth T. Montgomery.

**Project administration:** Seth Zissette, Thesla Palanee-Phillips, Florence Mathebula, Siyanda Tenza, Miriam Hartmann.

**Supervision:** Elizabeth E. Tolley, Thesla Palanee-Phillips, Miriam Hartmann.

**Validation:** Andres Martinez.

**Visualization:** Seth Zissette, Andres Martinez.

**Writing – original draft:** Elizabeth E. Tolley.

**Writing – review & editing:** Seth Zissette, Andres Martinez, Thesla Palanee-Phillips, Florence Mathebula, Siyanda Tenza, Miriam Hartmann, Elizabeth T. Montgomery.

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
