## [Decision Letter · Decision Letter 0]

27 Aug 2020

PONE-D-20-18744

Development and initial validation of a simple tool to screen for partner support or opposition to HIV prevention product use

PLOS ONE

Dear Dr. Tolley,

Thank you for submitting your manuscript to PLOS ONE. After careful consideration, we feel that it has merit but does not fully meet PLOS ONE’s publication criteria as it currently stands. Therefore, we invite you to submit a revised version of the manuscript that addresses the points raised during the review process.

I sent the manuscript to two experts in the field, who highlighted both strengths and weaknesses of the current version of the paper. As such, I invite the authors to submit a substantially revised version of the paper that addresses all concerns and responds in a point-by-point fashion to each.

We look forward to receiving your revised manuscript.

Kind regards,

H. Jonathon Rendina, PhD, MPH

Academic Editor

PLOS ONE

Journal Requirements:

2. Please ensure that you refer to Figure 1 in your text as, if accepted, production will need this reference to link the reader to the figure.

Reviewers' comments:

Reviewer's Responses to Questions

**Comments to the Author**

1. Is the manuscript technically sound, and do the data support the conclusions?

Reviewer #1: Yes

Reviewer #2: Yes

2. Has the statistical analysis been performed appropriately and rigorously? 

Reviewer #1: Yes

Reviewer #2: Yes

3. Have the authors made all data underlying the findings in their manuscript fully available?

Reviewer #1: Yes

Reviewer #2: No

4. Is the manuscript presented in an intelligible fashion and written in standard English?

Reviewer #1: Yes

Reviewer #2: Yes

5. Review Comments to the Author

Reviewer #1: Tolley et al. submitted an impactful paper developing and testing a screening tool for partner support in the context of HIV prevention among women in Sub-Sahara Africa. The developed scales could be (and are currently) used in survey research and clinical trials. The scales were developed with good methodology, including the use of cognitive interview procedures and item refinement. This is a well-written paper and I only have minor points to be considered in revision.

1. Based on the Methods described, it’s unclear if the authors conducted one large EFA with 127 items or if they conducted EFAs for each hypothesized construct.

2. It would be helpful if the authors reported whether each subscale was unidimensional (e.g., if they did not include any cross-loading and each construct only measured one theme). I raise this thought because of the relatively low factor loadings at times (0.40-0.50), which makes me think each construct could have subconstructs. This would be visible with subsequent factor analysis of only the items retained for each scale.

3. Some of the constructs have a lot of items (e.g., 13 for traditional values). This could artificially inflate the internal reliability. The authors could consider reducing the number of items using item-rest correlations (i.e., does reliability increase by removal of one item at a time?). Shorter scales might help with better integration into the clinical setting (and shorter surveys in research).

Reviewer #2: The manuscript "Development and initial validation of a simple tool to screen for partner support or opposition to HIV prevention product use" describes a scale development and validation study to develop a screening tool to assess potential facilitators and barriers for women's use of products to prevent HIV. The strengths of the manuscript include its thorough methodology and use of qualitative data to validate the quantitative measure. My answer to Question 3 regarding making data available is related to the fact that it seems as though some of the supplemental material referred to in the manuscript was not included in the submission (see comment 11 below). My more specific comments for ways that the authors can further strengthen the manuscript are as follows:

1. The transition from the Introduction to the Methods seems a little abrupt. I think it would be improved by ending your introduction with a paragraph highlighting the need for/potential purpose of a screening tool, and how this study will address that need.

2. Please provide more detail about the sample of 25 women with whom cognitive interviews were conducted (line 125). How were these women recruited? Did they meet the same eligibility criteria used for the cross-sectional survey (described lines 131-135)?

3. Please clarify whether all survey items were in English, and how responses may have differed for those for whom English is not their first or preferred language (e.g., those who preferred to complete their consent form in Zulu).

4. Please provide more explanation as to why items with a mean score below 1.4 or above 5.8 were eliminated. It occurs to me that these items could be viewed as valid and assessing the lowest and highest levels of the construct of interest (or reflect floor and ceiling effects seen in general, as described in the Discusssion), so more clarification about the reasoning around this decision would be helpful.

5. Please clarify what is meant by the sentence “All coders similarly applied the full codebook, as well as specific codes to text segments” (line 181-182). I’m not sure what is meant by the “full codebook” versus “specific codes.”

6. Regarding the questions about condom use and willingness to have sex at most recent sex act, I wonder about the potential limitations of these self-report data. Considering the potential power dynamics in these relationships, and that the majority of the women reported that they received financial or material support from their partners, it seems possible that social desirability and/or concerns about confidentiality could have influenced these responses. I had the same thought about this as a possible explanation for the floor/ceiling effects found for the Partner Resistance to HIV Prevention and HIV Prevention Readiness scales, which could possibly be due to participants answering in a socially desirable way related to their involvement in HIV prevention research.

7. It appears as though there is a formatting error in line 260, and Figure 1 is not included in the body of the manuscript.

8. Were any items added or removed based on their match (or lack thereof) with the qualitative text data? The description of this component of the analyses made it seems as though items would be revised based on these data, but there is no mention of this in the results section.

9. Please clarify if the version of HEART included in the CHARISMA pilot intervention included the HIV Prevention Readiness scale. If it was included, please justify the reasoning, given its poor performance in this analysis. Are there any plans to revise the scale?

10. I agree with the author’s suggestion that including items that address product-specific issues or concerns (e.g., taking a daily PrEP pill, concerns about comfort or sex with a vaginal ring) could strengthen the HIV Prevention Readiness scale.

11. It seems like there may be some supplemental materials that were left out of the manuscript submission. For example, I don’t see the equivalent of Tables 5, 6, or 7 for the other HEART scales.

6. PLOS authors have the option to publish the peer review history of their article (what does this mean?). If published, this will include your full peer review and any attached files.

Reviewer #1: No

Reviewer #2: No

---

## [Author Response · Author response to Decision Letter 0]

15 Oct 2020

Reviewer #1: Tolley et al. submitted an impactful paper developing and testing a screening tool for partner support in the context of HIV prevention among women in Sub-Sahara Africa. The developed scales could be (and are currently) used in survey research and clinical trials. The scales were developed with good methodology, including the use of cognitive interview procedures and item refinement. This is a well-written paper and I only have minor points to be considered in revision.

1. Based on the Methods described, it’s unclear if the authors conducted one large EFA with 127 items or if they conducted EFAs for each hypothesized construct.

Response: Thank you for this question. We have clarified in the manuscript, abstract and methods section, that it was one large EFA.

2. It would be helpful if the authors reported whether each subscale was unidimensional (e.g., if they did not include any cross-loading and each construct only measured one theme). I raise this thought because of the relatively low factor loadings at times (0.40-0.50), which makes me think each construct could have subconstructs. This would be visible with subsequent factor analysis of only the items retained for each scale.

Response: We have clarified that, by using a Promax rotation, we allowed items to cross-load on multiple factors. However, we did remove any items that did not load mainly on one factor. (If similar loading, we removed the item.) We have also clarified that one of the factors in our final solution was subsequently found to have subconstructs, which we utilized in the final tool. 

3. Some of the constructs have a lot of items (e.g., 13 for traditional values). This could artificially inflate the internal reliability. The authors could consider reducing the number of items using item-rest correlations (i.e., does reliability increase by removal of one item at a time?). Shorter scales might help with better integration into the clinical setting (and shorter surveys in research).

Response: We agree with the reviewer about the potential for larger number of items to artificially inflate the internal reliability of some scales. Because our initial scale development efforts were based on a cross-sectional survey (in addition to cognitive interviews), we did not want to be too restrictive in limiting the numbers of items or of scales that could be further evaluated in the pilot phase – and ultimately in the RCT phase of the project. In fact, in subsequent phases of this project, we have examined ways to shorten the HEART, including reducing the numbers of items in several scales and prioritization of HEART subscales for use in clinic settings. However, this will be presented in a subsequent paper. 

Reviewer #2: The manuscript “Development and initial validation of a simple tool to screen for partner support or opposition to HIV prevention product use” describes a scale development and validation study to develop a screening tool to assess potential facilitators and barriers for women’s use of products to prevent HIV. The strengths of the manuscript include its thorough methodology and use of qualitative data to validate the quantitative measure. My answer to Question 3 regarding making data available is related to the fact that it seems as though some of the supplemental material referred to in the manuscript was not included in the submission (see comment 11 below). 

Response: We apologize for this oversight. We have added/uploaded supplemental tables with illustrate quotes and our predictions for the other HEART subscales. We have also added supplemental tables with predictive text frequencies and comparison of our predictions for the other subscales. 

My more specific comments for ways that the authors can further strengthen the manuscript are as follows:

1. The transition from the Introduction to the Methods seems a little abrupt. I think it would be improved by ending your introduction with a paragraph highlighting the need for/potential purpose of a screening tool, and how this study will address that need.

Response: Thank you for this comment. We have added a short paragraph to make this transition more apparent. (Lines 85-89 on page 5). 

2. Please provide more detail about the sample of 25 women with whom cognitive interviews were conducted (line 125). How were these women recruited? Did they meet the same eligibility criteria used for the cross-sectional survey (described lines 131-135)?

Response: Yes, women were recruited through the same clinic and community mechanisms as women in the survey. Eligibility was the same as the survey. Because our analyses for this paper focused primarily on the survey, we didn’t want to go into too much detail for the cognitive interviews. However, we have tried to add some additional detail. 

3. Please clarify whether all survey items were in English, and how responses may have differed for those for whom English is not their first or preferred language (e.g., those who preferred to complete their consent form in Zulu).

Response: We conducted separate cognitive interviews in English and Zulu, assessing comprehension, relevance and requesting suggested rewording in both languages. We have added this information into the description. 

4. Please provide more explanation as to why items with a mean score below 1.4 or above 5.8 were eliminated. It occurs to me that these items could be viewed as valid and assessing the lowest and highest levels of the construct of interest (or reflect floor and ceiling effects seen in general, as described in the Discussion), so more clarification about the reasoning around this decision would be helpful.

Response: We compared the mean scores and standard distributions, modes and response ranges of all items across the entire sample (n=309) and by trial-experience subgroups (former trial participant and trial naïve). Many items showed skewed distributions in one of the subgroups, but only 4 items were highly skewed across the entire sample. We decided to remove these items prior to factor analysis as we concluded that the items provided virtually no variability through which we might differentiate between respondents. In all cases, similar items were retained in the item pool that elicited a wider range of responses. 

5. Please clarify what is meant by the sentence “All coders similarly applied the full codebook, as well as specific codes to text segments” (line 181-182). I’m not sure what is meant by the “full codebook” versus “specific codes.”

Response: Thank you for this query. We agree that the text was not very clear and have edited the text to make this description clearer (pages 187-190 on page 9). Essentially, the coders examined whether they had applied both the 5 HEART construct/basic codes (full) and the sub-codes indicating which of 3 levels was predicted by the text in the same way.

6. Regarding the questions about condom use and willingness to have sex at most recent sex act, I wonder about the potential limitations of these self-report data. Considering the potential power dynamics in these relationships, and that the majority of the women reported that they received financial or material support from their partners, it seems possible that social desirability and/or concerns about confidentiality could have influenced these responses. I had the same thought about this as a possible explanation for the floor/ceiling effects found for the Partner Resistance to HIV Prevention and HIV Prevention Readiness scales, which could possibly be due to participants answering in a socially desirable way related to their involvement in HIV prevention research.

Response: Thank you. You make an excellent point. In fact, concerns about social desirability bias in relation to direct questions about potentially unacceptable behaviors – IPV or non-adherence, is part of the justification for using a scale approach, allowing participants to provide more nuanced indications of their own or their partners behaviors. We have incorporated additional language about social desirability bias into the discussion section. 

7. It appears as though there is a formatting error in line 260, and Figure 1 is not included in the body of the manuscript.

Response: We had to upload this figure separately, so perhaps somehow it did not get included in the review. We have uploaded again. However, we believe it will be a separate document that will only be inserted at the time of (hopefully) publication. 

8. Were any items added or removed based on their match (or lack thereof) with the qualitative text data? The description of this component of the analyses made it seems as though items would be revised based on these data, but there is no mention of this in the results section.

Response: We did not remove any items based on this analysis. However, we did add an additional five “test” items to the HIV Prevention Readiness scale during the Expansion/RCT phase. These new items were based on the qualitative test data.

9. Please clarify if the version of HEART included in the CHARISMA pilot intervention included the HIV Prevention Readiness scale. If it was included, please justify the reasoning, given its poor performance in this analysis. Are there any plans to revise the scale?

Response: We included the HPR scale in the pilot, because we wanted to obtain additional information on this subscale. Given that the majority of participants in our formative research had not previously participated in clinical trials, we assumed that pilot participants might respond to these items differently. Unfortunately, the additional HPR items that emerged from our qualitative analysis were not identified early enough to be a part of the pilot. We did include these new items in the final Expansion/RCT study and they did improve the HPR scale. However, we will present and discuss this in a manuscript that focuses on the final phase of the project.

10. I agree with the author’s suggestion that including items that address product-specific issues or concerns (e.g., taking a daily PrEP pill, concerns about comfort or sex with a vaginal ring) could strengthen the HIV Prevention Readiness scale.

Response: As described in our response above, this scale has now been strengthened, but it continues to be less reliable than the other measures. This is likely because the scale was more generic, and not related to a specific product. 

11. It seems like there may be some supplemental materials that were left out of the manuscript submission. For example, I don’t see the equivalent of Tables 5, 6, or 7 for the other HEART scales.

Response: Thank you for pointing out this omission. We have included them now in the supplemental materials as tables S2-S4.

---

## [Decision Letter · Decision Letter 1]

11 Nov 2020

Development and initial validation of a simple tool to screen for partner support or opposition to HIV prevention product use

PONE-D-20-18744R1

Dear Dr. Tolley,

We’re pleased to inform you that your manuscript has been judged scientifically suitable for publication and will be formally accepted for publication once it meets all outstanding technical requirements.

Kind regards,

H. Jonathon Rendina, PhD, MPH

Academic Editor

PLOS ONE

Additional Editor Comments (optional):

Reviewers' comments:

Reviewer's Responses to Questions

**Comments to the Author**

1. If the authors have adequately addressed your comments raised in a previous round of review and you feel that this manuscript is now acceptable for publication, you may indicate that here to bypass the “Comments to the Author” section, enter your conflict of interest statement in the “Confidential to Editor” section, and submit your "Accept" recommendation.

Reviewer #1: All comments have been addressed

Reviewer #2: All comments have been addressed

2. Is the manuscript technically sound, and do the data support the conclusions?

Reviewer #1: Yes

Reviewer #2: Yes

3. Has the statistical analysis been performed appropriately and rigorously? 

Reviewer #1: Yes

Reviewer #2: Yes

4. Have the authors made all data underlying the findings in their manuscript fully available?

Reviewer #1: Yes

Reviewer #2: Yes

5. Is the manuscript presented in an intelligible fashion and written in standard English?

Reviewer #1: Yes

Reviewer #2: Yes

6. Review Comments to the Author

Reviewer #1: (No Response)

Reviewer #2: (No Response)

7. PLOS authors have the option to publish the peer review history of their article (what does this mean?). If published, this will include your full peer review and any attached files.

Reviewer #1: No

Reviewer #2: No

---

## [Editor Report · Acceptance letter]

11 Dec 2020

PONE-D-20-18744R1 

Development and initial validation of a simple tool to screen for partner support or opposition to HIV prevention product use 

Dear Dr. Tolley:

I'm pleased to inform you that your manuscript has been deemed suitable for publication in PLOS ONE. Congratulations! Your manuscript is now with our production department. 

Kind regards, 

on behalf of

Dr. H. Jonathon Rendina 

Academic Editor

PLOS ONE